# Adsorption Characteristics and Mechanisms of Fe-Mn Oxide Modified Biochar for Pb(II) in Wastewater

**DOI:** 10.3390/ijerph19148420

**Published:** 2022-07-10

**Authors:** Shang-Feng Tang, Hang Zhou, Wen-Tao Tan, Jun-Guo Huang, Peng Zeng, Jiao-Feng Gu, Bo-Han Liao

**Affiliations:** 1College of Environment Science and Engineering, Central South University of Forestry and Technology, Changsha 410004, China; tsf0515@163.com (S.-F.T.); tanvventao@163.com (W.-T.T.); flj020724@163.com (J.-G.H.); zengzengpp@foxmail.com (P.Z.); gujiaofeng@csuft.edu.cn (J.-F.G.); liaobh1020@163.com (B.-H.L.); 2Hunan Engineering Laboratory for Control of Rice Quality and Safety, Changsha 410004, China

**Keywords:** Fe-Mn oxide composite, biochar, wastewater, lead, adsorption mechanism

## Abstract

This study prepared iron-manganese oxide-modified biochar (FM-BC) by impregnating rice straw biochar (BC) with a mixed solution of ferric nitrate and potassium permanganate. The effects of pH, FM-BC dosage, interference of coexisting ions, adsorption time, incipient Pb(II) concentration, and temperature on the adsorption of Pb(II) by FM-BC were investigated. Moreover, the Pb(II) adsorption mechanism of FM-BC was analyzed using a series of characterization techniques. The results showed that the Fe-Mn oxide composite modification significantly promoted the physical and chemical functions of the biochar surface and the adsorption capacity of Pb(II). The specific surface area of FM-BC was 18.20 times larger than that of BC, and the maximum Pb(II) adsorption capacity reached 165.88 mg/g. Adsorption kinetic tests showed that the adsorption of Pb(II) by FM-BC was based on the pseudo-second-order kinetic model, which indicated that the adsorption process was mainly governed by chemical adsorption. The isothermal adsorption of Pb(II) by FM-BC conformed to the Langmuir model, indicating that the adsorption process was spontaneous and endothermic. Characterization analyses (Fourier transform infrared spectroscopy and X-ray photoelectron spectroscopy) showed that the adsorption mechanism of Pb(II) by FM-BC was mainly via electrostatic adsorption, chemical precipitation, complexation, ion exchange, and the transformation of Mn_2_O_3_ into MnO_2_. Therefore, FM-BC is a promising adsorbent for Pb(II) removal from wastewater.

## 1. Introduction

Lead (Pb) is a non-biodegradable, toxic element that bioaccumulates in the environment [1]. The mining and smelting, paint and coating, electroplating, and chemical industries are the main sources of Pb in aquatic environments [2]. Pb-polluted water presents a great danger to the ecological environment and threatens human health through the food chain [3,4]. Therefore, there is an urgent need to treat wastewater containing Pb and reduce the discharge of Pb into the environment in a high-efficiency, low-carbon, and environmentally friendly manner.

Recently, ion exchange [5], chemical precipitation [6], electrocoagulation [7], and adsorption [8] have been used to remove heavy metals from wastewater. Among these, adsorption is widely used because of its advantages of simple operation, high efficiency, and environmental friendliness, especially in the deep treatment of heavy metals in wastewater [9]. Biochar has a complex void structure, surface oxygen-containing functional group, and a large specific surface area [10,11], and is considered to be a valuable adsorbent for the removal of heavy metals from wastewater [12,13]. Biochar has a wide range of raw materials, such as rice straw [14], rapeseed stem [15] and corn straw [16], which are agricultural by-products. Rapid pyrolysis technology is a common method of preparing biochar from agricultural waste. Raw materials can be pyrolyzed under anoxic and at a temperature range of 300–1000 °C conditions to obtain raw biochar [17,18]. However, the surface of biochar has many negative charges and a low content of functional groups and its adsorption capacity for heavy metals is limited [19]. Researchers have enhanced the adsorption performance of biochar by UV radiation modification, acid-base modification, functional group loading, and other modification methods. Studies have shown that the modification of biochar with metal or metal oxides can significantly boost the specific surface area and microporous pore size, boost the load amount of functional groups, such as hydroxyl and carboxyl groups, and significantly improve the adsorption performance of biochar [20]. Fe and Mn oxides can accelerate electron transfer and improve the oxidation process and have been widely used for the surface modification of raw materials [17]. Fe oxide has a significant specific surface area, strong affinity, and a tendency toward Pb(II) [21]. Wang et al. discovered that the maximum adsorption capacity of Fe_3_O_4_ modified biochar for Pb(II) was 110 mg/g, compared to 77.5 mg/g for the original biochar. Zhao et al. indicated that hematite-modified biochar had an outstanding adsorption capacity for Pb(II) in water, which was 41.75% higher than the unmodified biochar. Mn oxide has strong oxidation characteristics, excellent adsorption performance, and can form stable complexes with Pb(II) [22,23]. Faheem et al. [24] found that when the coverage of manganese oxide on rice husk biochar was 40%, the specific surface area of the modified manganese oxide biochar was 1.45 times larger than that of the original biochar and the maximum adsorption capacity for Pb(II) was 86.5 mg/g. Wang et al. showed that the strong interaction between birnessite particles and Pb(II) could strengthen the adsorption process and that the removal effect of manganese oxide-loaded biochar on Pb(II) in water (47.05 mg/g) was better than that of the original biochar (2.35 mg/g).

There is a synergistic effect between Fe and Mn metal oxides and the combination of Fe and Mn oxide modification can significantly improve the adsorption performance of biochar compared to the single-oxide modification [25]. Therefore, we believe that the Fe and Mn oxide composite-modified materials have a better adsorption capacity for Pb(II) in wastewater. However, little research has been conducted on the adsorption features and mechanism of Pb(II) in water by the combined modification of biochar with Fe and Mn oxides. In this study, rice straw biochar (BC) was used as the raw material and Fe-Mn oxide biochar (FM-BC) was prepared by impregnating it with a combination of Fe and Mn oxides. The major objectives of the present study were: (1) to study the adsorption performance and adsorption characteristics of FM-BC for Pb(II) in wastewater; (2) to investigate the influences of pH, addition dosage, coexisting ions, adsorption time, incipient concentration, and temperature on the Pb(II) adsorption capacities of FM-BC; (3) to explore the adsorption mechanism of Pb(II) in wastewater by FM-BC.

## 2. Materials and Methods

### 2.1. FM-BC Preparation

Rice straw was obtained from a paddy field in Liuyang City, Hunan Province (China), and was washed and dried in an electrothermal blowing drying oven (101-1AB, Tianjin Taiste, Tianjin, CHN) to a constant weight before being crushed by a 100-mesh sieve to make straw powder. The straw powder was heated in an anaerobic environment at 300 °C for 4 h to form biochar. Then, the biochar was cooled down to room temperature, washed with deionized water, and desiccated at 70 °C to obtain the original biochar of straw (BC).

We prepared biochars with nine different molar ratios of Fe/Mn and, through a pre-test, selected the Fe-Mn oxide biochar with the strongest adsorption capacity for Pb(II), Fe:Mn = 2:5. The composite modification of BC with Fe-Mn oxides was conducted by the impregnation method, and BC (5.0 g) was weighed and infiltrated into a mixed solution of 40 mL of 0.10 mol/L Fe(NO_3_)_3_ and 40 mL of 0.25 mol/L KMnO_4_. The mixture was stirred in a magnetic stirrer for 2 h, then placed in a water bath at 95 °C for 22 h, dried, and subjected to anaerobic pyrolysis at 300 °C for 0.5 h. The mixture was then cooled to room temperature, washed with deionized water, and dried in a dryer at 70 °C to a constant weight to obtain Fe-Mn oxide biochar (FM-BC).

### 2.2. Adsorption Experiments

These experiments examined the impact of pH (2.0, 3.0, 4.0, 5.0 and 6.0), FM-BC dosage (0.5, 1, 2, 4 and 8 g/L), single coexisting cations (Ca(II), Mg(II), Cu(II),and Zn(II)), adsorption time (0–720 min), incipient Pb(II) concentration (50, 100, 200 and 400 mg/L), and reaction temperature (288, 298, 308 and 318 K) on the FM-BC adsorption performance for Pb(II). In addition, the regeneration ability of FM-BC was also studied. The supernatant was filtered through a microporous membrane (0.45 μm) and stored in plastic polyethylene bottles for testing. All treatments were repeated three times. The adsorption process scheme is detailed in the Appendix A.

### 2.3. Characterization Analysis Method

Samples of BC, FM-BC, and FM-BC after adsorption (FM-BC-Pb) were characterized and analyzed. The microstructure and surface properties of the samples were analyzed by scanning electron microscopy (SEM, SIGMA 300, Oberkohen, GER) and the specific surface area of the samples was analyzed by Brunauer–Emmett–Teller method (BET) with automatic specific surface area analyzer (ASAP 2020 HD88, AUTOSORB IQ, Atlanta, GA, USA). The chemical composition and valence state of elements were examined using X-ray photoelectron spectroscopy (XPS, Thermo Scientific K-Alpha, Waltham, MA, USA) and the functional group structures of the samples were analyzed using Fourier transform infrared spectroscopy (FTIR, Thermo IN10, Waltham, MA, USA) in the absorption mode of 4000–400 cm^−1^.

### 2.4. Sample Analysis Method and Quality Control

The Pb(II) content in the filtrate was examined using ICP-AES (Thermo Fisher, ICAP-6300, Portland, OR, USA). The detection limit for Pb was 0.020 mg/L. The standard addition recovery method was used for quality control, and the recovery rate of Pb was 99.6–104.0%.

### 2.5. Data and Statistics Analysis

Pb(II) removal rate and adsorption capacity by FM-BC were evaluated, and the adsorption kinetics were investigated using pseudo-first-order and pseudo-second-order kinetics. To assess the isothermal adsorption of the experimental data and explore the adsorption thermodynamics, the Langmuir and Freundlich equations were utilized. Detailed data processing is described in the Appendix A. The data were processed and analyzed using Microsoft Office Excel and SPSS statistics V22, analyzed using the “post hoc” Duncan test (*p* < 0.05) and the analysis of variance (ANOVA) univariate, and mapped with OriginPro 2019b.

## 3. Results and Discussion

### 3.1. The Peculiarities of the Adsorbent Structure

#### 3.1.1. SEM and BET Analysis

The morphological characteristics and microstructures of BC, FM-BC, and FM-BC-Pb are displayed in Figure 1. The pore structure of BC was small and with few pores, and its surface was relatively smooth (Figure 1a). The modified FM-BC had pores all over the biochar surface with a larger pore structure, and more pores and spherical Fe-Mn binary oxide fine particles were attached to the surface. Furthermore, the elemental peaks of Mn and Fe in the EDS elemental analysis increased, confirming that Mn and Fe were effectively loaded onto the BC surface (Figure 1b). Moreover, the nanoparticles of metal (Ca(II), K(I)) produced during pyrolysis were also visible from the FM-BC surface, which was more favorable for ion exchange between FM-BC and Pb(II) [26]. Following the Pb(II) adsorption, the surface and pores of FM-BC were filled with a large number of granular Pb compounds (Figure 1c), indicating that FM-BC could efficiently adsorb Pb(II). As shown in Table 1, compared to BC, the specific surface area of FM-BC (13.726 m^2^/g) was 18.20 times higher, and the total pore volume was 15.00 times higher. This might be due to pyrolysis promoting the thermochemical decomposition of biomass and changing the pore structure. Fe-Mn oxide affected the pore structure of BC and optimized the surface properties of the biochar [27], which was beneficial for the adsorption of Pb(II) by FM-BC in wastewater. After Pb(II) adsorption, the specific surface area of FM-BC reached 26.434 m^2^/g. A large amount of Pb(II) was adsorbed and attached to the FM-BC surface and pores, which increased the specific surface area but reduced the average pore size (Table 1). In addition, the external surface area, micropore-specific surface area, and micropore volume of FM-BC were significantly higher (30.45, 6.34 and 8.00 times, respectively), than those of BC. This shows that the adsorption size of FM-BC increased, making it superior for the Pb(II) adsorption.

#### 3.1.2. XPS Analysis

The adsorption process of Pb by FM-BC was further investigated using XPS. As may be seen in Figure 2a, the major XPS peaks of FM-BC mainly included Fe 2p (711 eV), Mn 2p (642 eV), O 1s (530 eV), C 1s (286 eV), and Pb 4f (139 eV) peaks. The C 1s peak (Figure 2b) are divided into three peaks: C-C/C=C (284.7 eV), C-O-C (285.7 eV), and O-C=O (289.0 eV). C-C/C=C accounted for the largest proportion of BC and FM-BC-Pb (50.63% and 50.21%, respectively), and C-O-C accounted for the largest proportion of FM-BC (48.28%). This indicates that the C-C/C=C bond of biochar is oxidized by Fe-Mn oxide and then forms more C-O-C and O-C=O oxygen-containing functional groups with excellent properties [28], which is more conducive to the complexation of FM-BC with Pb(II). The Fe 2p spectrum (Figure 2c) showed that the three peaks with binding energies of 711.0, 713.1, and 714.8 eV belong to Fe_2_O_3_, FeOOH, and Fe(III), respectively. Fe_2_O_3_ accounted for the main proportion of both FM-BC and FM-BC-Pb, but its percentage decreased from 48.60% in FM-BC to 45.67% in FM-BC-Pb. The proportion of Fe(III) also decreased slightly, whereas the percentage of FeOOH increased (from 34.95% to 43.70%). The characteristic peak of Fe 2p had no obvious displacement, signifying that the Fe valence is fixed and does not change after the Pb(II) adsorption [22]. The Mn2p spectrum (Figure 2d) showed that the three peaks with binding energies of 641.2, 642.0, and 643.5 eV belong to MnO, Mn_2_O_3,_ and MnO_2_, respectively. Following the adsorption of Pb(II) by FM-BC, the proportion of Mn_2_O_3_ decreased from 63.43% to 34.18%, whereas the proportion of MnO_2_ increased from 23.76% to 42.38%, indicating that Mn_2_O_3_ participated in the adsorption process and Mn(III) underwent oxidation to form Mn(IV).

An obvious Pb 4f peak appeared in the XPS spectrum of FM-BC-Pb (Figure 2e). The characteristic peaks of Pb(OH)_2_, PbCO_3,_ and Pb(NO_3_)_2_ appeared at 138.7, 139.0 and 139.2 eV, respectively, of which PbCO_3_ accounted for the main proportion (40.02%). The proportion of Pb(NO_3_)_2_ accounted for 35.51%, which may have been caused by the FM-BC remaining on the material surface after adsorbing the solution with Pb(NO_3_)_2_ as a solute.

#### 3.1.3. FTIR Analysis

The functional groups of BC, FM-BC, and FM-BC-Pb were tested using FTIR (Figure 3). The wide peak of FM-BC at 3434 cm^−1^ was caused by the tensile and oscillation peaks of the -OH group of alcohol or phenol [29], that at 2922 cm^−1^ was prompted by the deformation oscillation of -CH_2_, that at 1625 cm^−1^ was prompted by the tensile oscillation of C=C, -C=O, those at 1385 and 1072 cm^−1^ were the absorption peaks of carboxyl O=C-O and C-O-C of cellulose [30], and that at 554 cm^−1^ was the absorption peaks of Fe-O and Mn-O [31]. The results showed that Fe-Mn oxide was effectively loaded on the biochar surface, which was consistent with the XPS results. FM-BC contained a variety of oxygen-containing functional groups, which are conducive to the Pb(II) adsorption.

### 3.2. Impacts of pH, Addition Dosage and Coexisting Ions on Pb(II) Adsorption by FM-BC

#### 3.2.1. The Effect of pH on the Adsorption of Pb(II) by FM-BC

Figure 4 showed that the Pb(II) adsorption capacity and removal rate in water by FM-BC were influenced by the pH of the aqueous solution. At pH 2.0, the FM-BC for the adsorption capacity was only 5.99 mg/g and the adsorption effect was the lowest. The reason was that the surface of FM-BC was positively charged and the electrostatic repulsive force limited the adsorption of Pb(II) in acidic environments [32] and accelerated the desorption rate of Pb(II) in the FM-BC [33]. When the solution’s pH value increased from 2.0 to 5.0, the reduction rate of Pb(II) increased rapidly from 6.33% to 95.20% and the adsorption capacity rose from 5.99 to 94.30 mg/g. The main reason was that H^+^ in the aqueous solution decreased significantly with the increasing pH; moreover, FM-BC had a large number of negative charges on the surface, which was beneficial for the adsorption of Pb(II) cations. When the pH of the aqueous solution was changed from 5.0 to 6.0, the contents of OH^-^ and CO_3_^2-^ in the solution increased and OH^-^ and Pb(II) formed Pb(OH)_2_ and Pb(OH)^+^ [34], which reduced the positive charge of Pb and the content of free moving Pb(II), weakened the electrostatic attraction between FM-BC and Pb(II), and caused a minor drop in the adsorption capacity of Pb(II) by FM-BC (2.77 mg/g). It further confirmed that the Pb(II) adsorption could be related to hydrogen bonding and ion exchange [35].

#### 3.2.2. The Effect of FM-BC Addition Dosage on Pb(II) Adsorption by FM-BC

The effects of different FM-BC addition dosages on Pb(II) adsorption in water can be seen in Figure 5. As the dosage of FM-BC increased from 0.5 to 8.0 g/L, the amount of Pb(II) adsorption gradually decreased, from 124.80–207.51 to 23.61–24.66 mg/g. However, the removal rate of Pb(II) increased from 31.50–52.38% to 95.34–99.59% (Appendix A). This is because the number of effective adsorption sites for Pb(II) in water increased with increasing FM-BC addition dosage. In contrast, the original content of Pb(II) in the solution was fixed (200 mg/L) and the molar amount of Pb(II) was constant, which resulted in the vacancy of some adsorption sites of FM-BC. As a result, FM-BC did not fully use all adsorption sites [36].

#### 3.2.3. The Effect of Coexisting Ions on Pb(II) Adsorption by FM-BC

The effects of coexisting divalent metal ions (Ca(II), Mg(II), Cu(II) and Zn(II)) on Pb(II) adsorption by FM-BC were investigated to represent the coexistence of other metal ions in wastewater. Figure 6 showed that the order of the effect of the four coexisting ions on Pb(II) adsorption by FM-BC was Cu(II) > Zn(II) > Ca(II) > Mg(II). The presence of Ca(II), Mg(II), and Zn(II) had little effect on the removal of Pb(II) by FM-BC. When the contents of Ca(II), Mg(II), and Zn(II) increased from 10 to 200 mg/L, the adsorption potential of FM-BC for Pb(II) decreased by only 1.61–10.01%, 1.47–8.60%, and 2.12–12.67%, respectively, indicating that Pb(II) had stronger competitiveness for adsorption sites than these coexisting ions. In general, a higher electronegativity value for metal ions results in a stronger attractive force with FM-BC, and a smaller radius of hydrated ions for metal ions results in a greater affinity with FM-BC [37,38]. The electronegativity values of Ca(II), Mg(II), Zn(II), and Pb(II) are 1.00, 1.31, 1.65, and 2.33, respectively, and the radius of the hydrated ions are 4.12, 4.28, 4.30, and 4.01 Å, respectively [39,40,41]. Unlike these three metal ions, Pb(II) has a higher electronegativity and a smaller hydrated ion radius. Therefore, when Ca(II), Mg(II), and Zn(II) coexisted, FM-BC exhibited better selectivity for the adsorption of Pb(II). However, the coexistence of Cu(II) significantly repressed the adsorption of Pb(II); when Cu(II) was 200 mg/L, the adsorption of Pb(II) by FM-BC was only 37.23 mg/g, decreased by 60.60%, and the adsorption amount of Cu(II) was 26.18 mg/g. This is because the electronegativity value of Cu(II) (1.90) was closer to that of Pb(II) (2.33) than those of Ca(II) (1.00), Mg(II) (1.31), and Zn(II) (1.65). Meanwhile, the hydrated ion radius of Cu(II) (4.19 Å) was similar to that of Pb(II) (4.01 Å) [39,40], and the hydrolysis constants of Pb(II) and Cu(II) were 7.7 [42]. According to ion exchange theory, the existence of Cu(II) had a great competitive impact on the Pb(II) adsorption by FM-BC, but the Pb(II) adsorption capacity by FM-BC was better than that of Cu(II). Therefore, the Cu content should be reduced as much as possible when applying the FM-BC to adsorb Pb(II) from wastewater to avoid its influence on Pb adsorption efficiency.

### 3.3. Adsorption Characteristics of FM-BC on Pb(II) in Water

#### 3.3.1. Adsorption Kinetics

The effects of initial content and adsorption time on Pb(II) adsorption by FM-BC are shown in Figure 7. FM-BC adsorbed Pb(II) faster and the maximum adsorption capacity could reach 68.25–99.66% of the equilibrium adsorption capacity when adsorbed for 5 min. This may be because the adsorption dynamic gradient between the FM-BC and Pb(II) solution interface was large and the adsorption locations on the surface of FM-BC were quickly filled with Pb(II). At low Pb(II) contents (50, 100 and 200 mg/L), the adsorption capacity curve of FM-BC for Pb(II) increased sharply with increasing adsorption time and reached equilibrium after 60 min. At this time, Pb(II) was completely adsorbed and the equilibrium adsorption capacities of FM-BC were 25.57, 50.47 and 94.55 mg/g for 50, 100 and 200 mg/L, respectively. Under the high content of Pb(II) (400 mg/L), the adsorption size curve of FM-BC for Pb(II) showed a slow upward trend with ascending adsorption time and did not reach equilibrium within 720 min. Pb(II) in the water body was not fully adsorbed, achieving a maximum adsorption size of 165.88 mg/g. This can be credited to the limited number of active adsorption locations on the surface of the FM-BC. With increasing Pb(II) content, the adsorption sites of FM-BC tended to saturate, which limited and slowed the adsorption [19].

According to the fitting data (Appendix A), Pb(II) adsorption by FM-BC conformed to the pseudo-second-order kinetic model and the *R^2^* was between 0.998–0.999, which was significantly higher than that of the pseudo-first-order kinetic model (0.656–0.975) (Appendix A). The pseudo-second-order kinetic model showed that the equilibrium adsorption size of FM-BC for Pb(II) was 166.67 mg/g, which was closer to the experimental value of 165.88 mg/g. In this study, the equilibrium adsorption size of FM-BC for Pb(II) was 166.67 mg/g, much higher than that of BC (70.33 mg/g). Furthermore, the maximum adsorption size of FM-BC for Pb(II) was significantly higher than other adsorbents stated in the recent literature (Table 2).

#### 3.3.2. Adsorption Isotherm

The adsorption isotherm of FM-BC for Pb(II) in water is displayed in Appendix A. The adsorption size of Pb(II) of FM-BC improved with increasing temperature (288–318 K). As the temperature amplified from 288 to 298 K, the increase in the adsorption size of FM-BC for Pb(II) was the largest and then slightly improved with further increase to 318 K. This showed that the adsorption process of FM-BC on Pb(II) was endothermic and the adsorption size of FM-BC on Pb(II) reached saturation at 298 K. The Freundlich and Langmuir models were utilized to fit the Pb(II) adsorption process by FM-BC (Appendix A). The outcomes disclosed that the adsorption process of FM-BC on Pb(II) was in harmony with the Langmuir model more than with the Freundlich model, with an *R^2^* between 0.996–0.999. The adsorption process of Pb(II) on FM-BC was speculated to be monolayer adsorption. At 288, 298, 308 and 318 K, the largest adsorption capacities of FM-BC calculated by the Langmuir model were 135.14, 147.06, 149.25, and 151.52 mg/g, respectively. At 298 K (room temperature), the adsorption capacities of Pb(II) by magnetic pine bark waste biochar and sludge ash in a past study were 29.699 and 25.0 mg/g, respectively, obviously inferior to that of FM-BC prepared in this study [47,48]. Therefore, FM-BC exhibits a stronger adsorption performance for Pb(II) than other materials. At the four different temperatures, the *R_L_* of FM-BC for Pb(II) adsorption was between 0–1, indicating that FM-BC was favorable for Pb(II) adsorption. These results indicated that the higher the temperature was, the closer the *R_L_* value was to 0 and the stronger the Pb(II) adsorption size of FM-BC. In conclusion, the adsorption of Pb(II) by FM-BC was monolayer, uniform, and heterogeneous.

#### 3.3.3. Adsorption Thermodynamics

To determine the spontaneity of the adsorption process, adsorption experiments of Pb(II) by FM-BC were carried out at four separate temperatures (288, 298, 308 and 318 K) and thermodynamic parameters were calculated (Appendix A). With increasing temperature, the Δ*G^θ^* value of FM-BC decreased from −7.91 to −12.07 kJ/mol. Δ*G^θ^* was negative, which indicated that the process of adsorption was spontaneous; the greater the temperature, the more conducive the FM-BC adsorption of Pb(II) and the stronger the spontaneity of the adsorption process. When Δ*H^θ^* > 0, it was further confirmed that Pb(II) absorbed by FM-BC was an endothermic process. Δ*S^θ^* > 0 showed that the disorder and randomness of molecules at the solid–liquid interface were amplified in the FM-BC adsorbing of Pb(II) and the Δ*S^θ^* value of FM-BC was 0.14 kJ/(mol·K) [49].

### 3.4. Adsorption Mechanism of Pb(II) by FM-BC

According to the XPS analysis (Figure 2), PbCO_3_ accounted for the main proportion, which indicated that chemical precipitation and complexation mainly occurred during the Pb(II) adsorption by FM-BC. The high content of Pb(NO_3_)_2_ indicated that physical adsorption was included in the sorption process. Moreover, a decrease in potassium ions in the EDX analysis (Figure 1) may indicate that potassium salts have washed out of the surface or pores and depending on the anions, Pb(II) probably formed more insoluble salt PbCO_3_.

According to the FTIR analysis (Figure 3), once the FM-BC adsorbed Pb(II), the intensities of the -OH, O=C-O, and C=C peaks of alcohols or phenols decreased, indicating that O=C-O, -OH, and C=C bonds might be involved in the adsorption process. The complexation of Pb(II) with ionized oxygen functional groups (-OH or -COO) or C=C (π-electron) bonds form a complex of Pb-O and COO-Pb [50]. In addition to that, Pb(II) can be adsorbed through an ion-exchange mechanism. as shown in the following equations [35]:2(R-COH) + Pb^2+^ → (R-CO)_2_Pb + 2H^+^.
2R-COOH + Pb^2+^ → (R-COO)_2_Pb + 2H^+^.

According to the findings of adsorption kinetics (Figure 7) and adsorption thermodynamics (Appendix A), the pseudo-second-order kinetic model displayed a more accurate description of the adsorption characteristics for Pb(II) on FM-BC, which indicated that the adsorption of Pb(II) on FM-BC was mainly governed by chemical adsorption [51]. In general, chemisorption was dominant when Δ*H^θ^* was between 20–200 kJ/mol [52]. Δ*H^θ^* of FM-BC was 32.15 kJ/mol, which indicated that chemical adsorption was the main adsorption process, agreeing with the pseudo-second-order kinetic fitting results.

## 4. Conclusions

FM-BC prepared by the composite modification of Fe and Mn oxides optimized the physical and chemical properties of the biochar surface. The specific surface area of FM-BC was 18.20 times higher than that of BC and the number of adsorption sites and pore structures was higher. The largest adsorption capacity of FM-BC for Pb(II) was 165.88 mg/g, achieved at 298 K. The presence of Cu(II) in the water had a significant competitive influence on the adsorption of Pb(II) by FM-BC. The adsorption of Pb(II) by FM-BC followed the pseudo-second-order kinetic and Langmuir models. The adsorption of Pb(II) by FM-BC is controlled by chemical adsorption, which is single-layer, uniform, and heterogeneous, and the adsorption process is spontaneous and endothermic. Through characterization and analysis, it was determined that Fe-Mn oxides participated in the adsorption of Pb(II). Fe_2_O_3_ and Fe(III) were transformed into FeOOH, while Mn_2_O_3_ was transformed into MnO_2_. The adsorption mechanism of FM-BC for Pb(II) mainly consisted of chemical precipitation, complexation, and ion exchange. In conclusion, Fe-Mn oxide modified biochar can effectively adsorb Pb(II), which is a promising biomass adsorption material for removing Pb(II) from wastewater.

## Figures and Tables

**Figure 1 ijerph-19-08420-f001:**
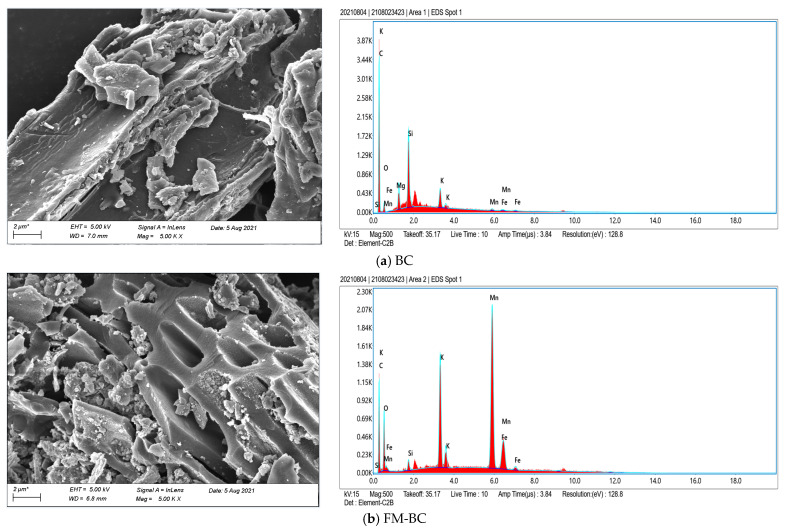
SEM-EDS images of (**a**) rice straw biochar (BC), (**b**) Fe-Mn oxide biochar (FM-BC), and (**c**) Fe-Mn oxide biochar after adsorption of Pb(II) (FM-BC-Pb).

**Figure 2 ijerph-19-08420-f002:**
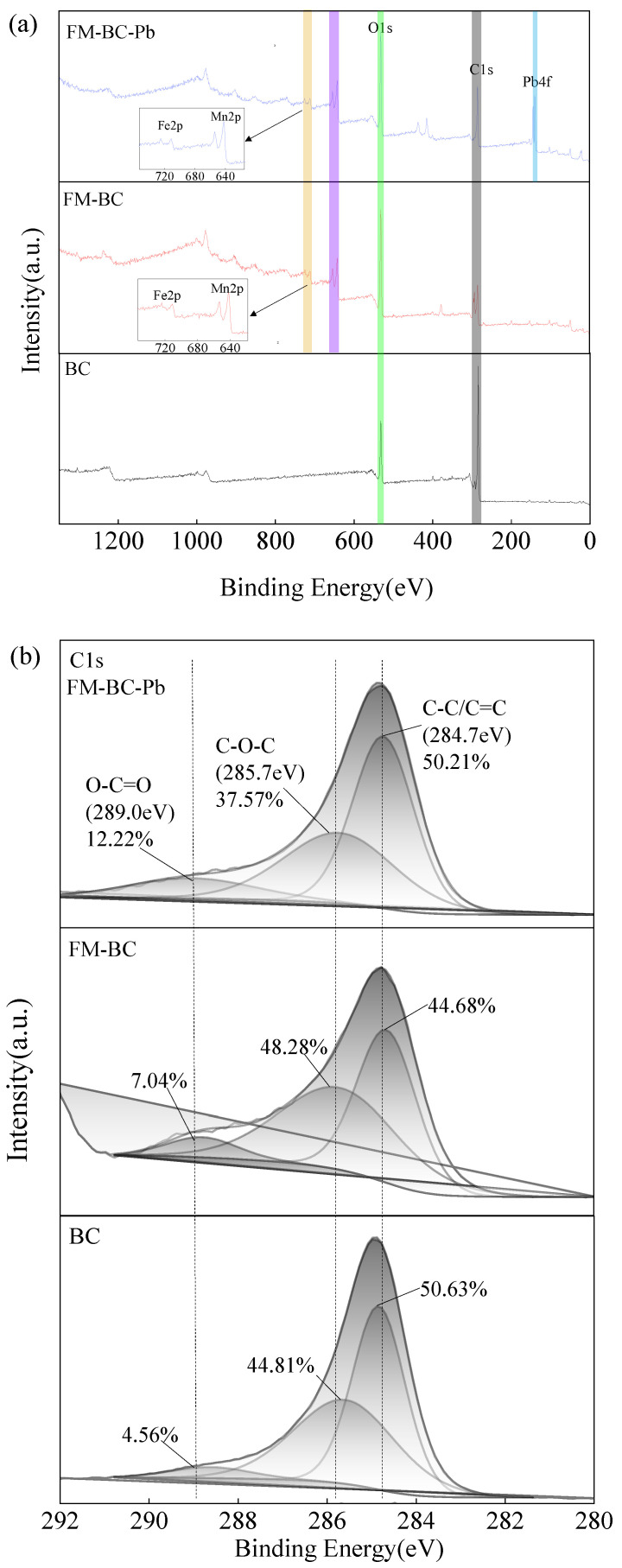
XPS spectrum of biochar (**a**) full scan spectrum (**b**) C 1s spectrum (**c**) Fe 2p spectrum (**d**) Mn 2p spectrum (**e**) Pb 4f spectrum.

**Figure 3 ijerph-19-08420-f003:**
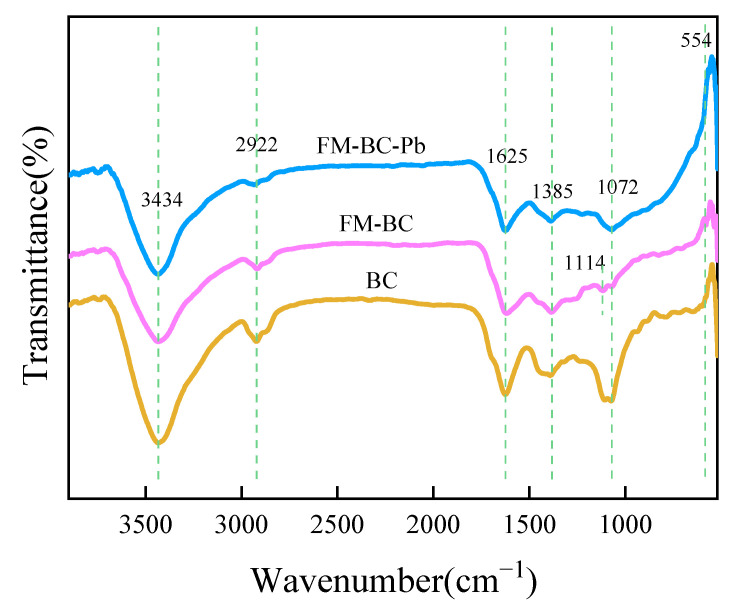
FTIR spectrums of biochar.

**Figure 4 ijerph-19-08420-f004:**
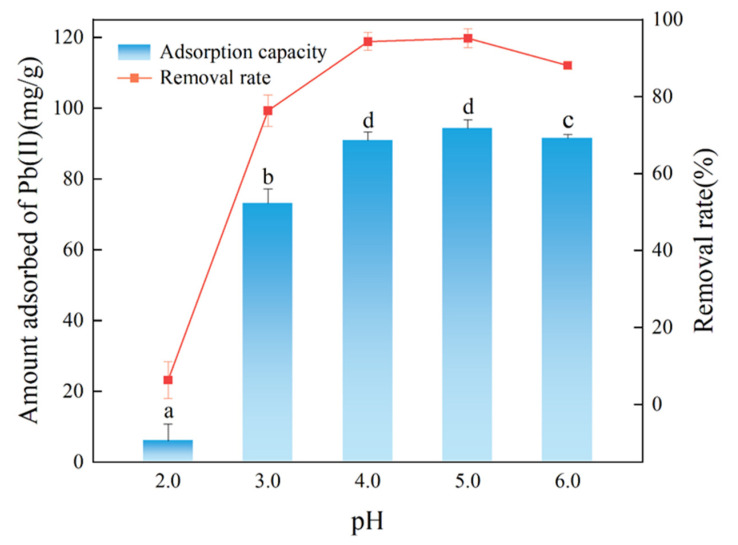
The effect of pH value on Pb(II) adsorption by FM-BC. Different letters indicate significant differences (*p* < 0.05).

**Figure 5 ijerph-19-08420-f005:**
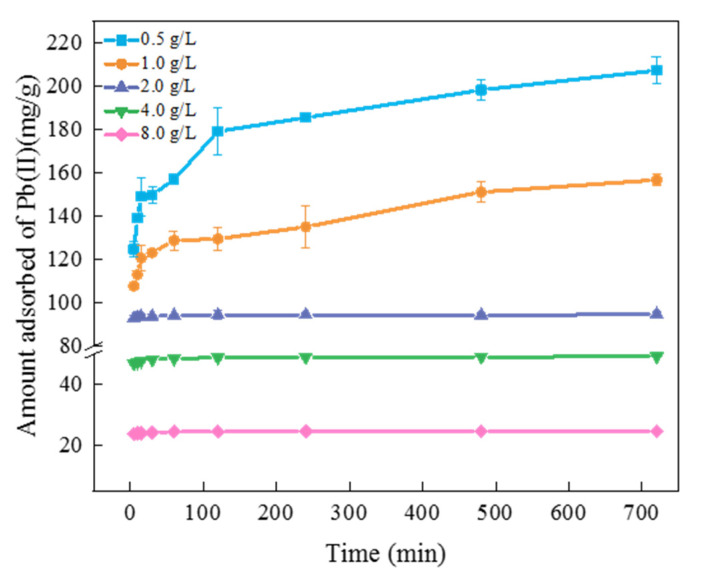
The effect of FM-BC addition on Pb(II) adsorption by FM-BC.

**Figure 6 ijerph-19-08420-f006:**
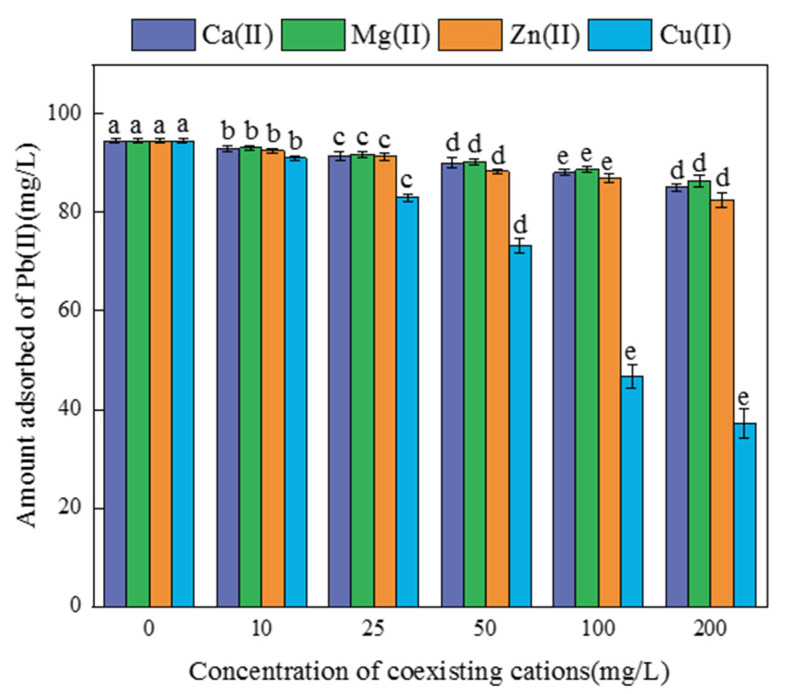
The effect of coexisting ions on Pb(II) adsorption by FM-BC. Different letters indicate significant differences (*p* < 0.05).

**Figure 7 ijerph-19-08420-f007:**
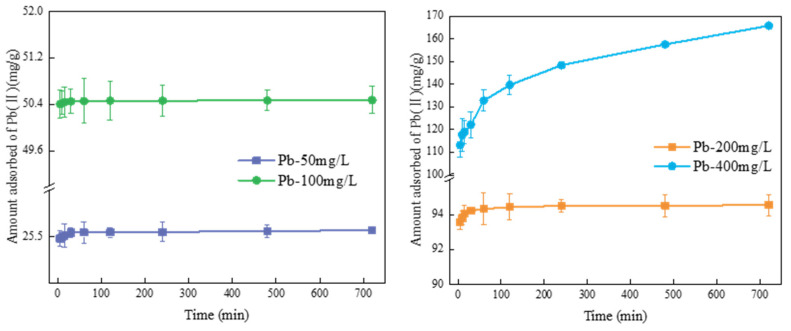
The effects of time and initial concentration on Pb(II) adsorption by FM-BC.

**Table 1 ijerph-19-08420-t001:** Surface property parameters of biochar.

Materials	Surface Area	External Surface Area	Micropore Specific Surface Area	Pore Diameter	Total Pore Volume	Micropore Volume
	(m^2^/g)	(m^2^/g)	(m^2^/g)	(nm)	(cm^3^/g)	(cm^3^/g)
BC	0.754	0.371	0.383	8.089	0.0002	0.0001
FM-BC	13.726	11.296	2.430	10.404	0.003	0.0008
FM-BC-Pb	26.434	19.884	6.550	7.008	0.006	0.002

**Table 2 ijerph-19-08420-t002:** Comparison of sorption capacity of adsorbents for Pb(II).

Adsorbent	Max. Pb(II) Capacity (mg/g)	References
Corn straw biochar	81.63	[43]
Shrimp shell biochar	94.16	[44]
Calcined mussel shell powder	102.04	[45]
CuFe_2_O_4_-loaded corncob biochar	132.10	[46]
BC	70.33	This study
FM-BC	165.88	

## Data Availability

The data for this study are available from the corresponding author via email: zhouhang4607@163.com.

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
