# Peer review of "Adsorption Characteristics and Mechanisms of Fe-Mn Oxide Modified Biochar for Pb(II) in Wastewater"

_ijerph, 2022, doi:10.3390/ijerph19148420_

Round 1

Reviewer 1 Report

The work is interesting, well-written, structured, and has all the details of synthesis, measurement, and experiment. But some points should be clarified:

1. The text contains typos or is poorly transformed into PDF, it should be checked.

2. Section 2.3 – “a Brunauer–Emmett– Teller analysis” – what does it mean? there is a misuse of terminology.

3. Section 2.3 –  “oxidation state of the samples” - there is a misuse of terminology.

4. Authors often use the phrase "reduction of Pb(II)", but this reduction has not been proven. Therefore it is better to write adsorption, uptake, or removal.

5. The authors compare the surface properties of the synthesized composite with other biochars obtained from different raw materials in [30] and [32]. This is not correct at all, but in particular, because you have modified your biochar.

6. The presentation of the material is non-standard; it may be at the discretion of the authors. But the sections 3.1 and 3.2 are devoted to adsorption which is difficult to explain because the clear structure of the material was not studied, and not understood by readers yet. As a rule, the articles demonstrate the peculiarities of the adsorbent structure firstly, then the adsorption is studied and the mechanism is considered. 

7. Section 3.3.1: An increase in the specific surface area value after adsorption, as well as a decrease in potassium ions in the EDX analysis may indicate that potassium salts have washed out of the surface or pores and depending on the anions, Pb(II) could form insoluble salts. For example, PbCO3.

8. And so there is still the question of the mechanism of adsorption, which may involve electrostatic attraction, hydrogen bonding, physical adsorption, ion exchange, and chemical precipitation (https://doi.org/10.1039/D0RA08055A) . This discussion needs to be expanded.

Author Response

Comment 1: The text contains typos or is poorly transformed into PDF, it should be checked.

Response:

Thank you for your helpful suggestions. We have checked the text for typos and format problems. Please see the revised manuscript.

Comment 2: Section 2.3 – “a Brunauer–Emmett– Teller analysis” – what does it mean? there is a misuse of terminology.

Response:

Thank you for your helpful suggestions. “a Brunauer–Emmett– Teller analysis” is the specific surface area analysis method, and we have rewritten this sentence. Please see our revised manuscript.

Section 2.3: The microstructure and surface properties of the samples were analyzed by scanning electron microscopy (SEM, SIGMA 300, GER) and the specific surface area of the samples were analyzed by Brunauer–Emmett–Teller method (BET) with automatic specific surface area analyzer (ASAP 2020 HD88, AUTOSORB IQ, USA).

Comment 3: Section 2.3 – “oxidation state of the samples” - there is a misuse of terminology.

Response:

Thank you for your helpful suggestions. We have rewritten the sentence. Please see our revised manuscript.

Section 2.3: The chemical composition and valence state of elements were examined using X-ray photoelectron spectroscopy (XPS, Thermo Scientific K-Alpha, USA).

Comment 4: Authors often use the phrase "reduction of Pb(II)", but this reduction has not been proven. Therefore it is better to write adsorption, uptake, or removal.

Response:

Thank you for your helpful suggestions. We have corrected the phrase "reduction of Pb(II)" in the text. Please see our revised manuscript.

Comment 5: The authors compare the surface properties of the synthesized composite with other biochars obtained from different raw materials in [30] and [32]. This is not correct at all, but in particular, because you have modified your biochar.

Response:

Thank you for the helpful comment. We have rewritten this section. Please see our revised manuscript.

Section 3.2.1: The main reason was that H+ in the aqueous solution decreased significantly with increasing pH; moreover, FM-BC had a large number of negative charges on the surface, which was beneficial for the adsorption of Pb(II) cations.

It further confirmed that the Pb(II) adsorption could be related to hydrogen bonding and ion exchange.

Comment 6: The presentation of the material is non-standard; it may be at the discretion of the authors. But the sections 3.1 and 3.2 are devoted to adsorption which is difficult to explain because the clear structure of the material was not studied, and not understood by readers yet. As a rule, the articles demonstrate the peculiarities of the adsorbent structure firstly, then the adsorption is studied and the mechanism is considered.

Response:

Thank you for the helpful comment. We changed the structure of the manuscript, first showed the peculiarities of the adsorbent structure, then studied the adsorption and considered the adsorption mechanism. Please see our revised manuscript.

3.1 The peculiarities of the adsorbent structure

3.1.1 SEM and BET analysis

3.1.2 XPS analysis

3.1.3 FTIR analysis

3.2 Impacts of pH, addition dosage, and coexisting ions on Pb(II) adsorption by FM-BC

3.2.1 The effect of pH on the adsorption of Pb(II) by FM-BC

3.2.2 The effect of FM-BC addition dosage on Pb(II) adsorption by FM-BC

3.2.3 The effect of coexisting ions on Pb(II) adsorption by FM-BC

3.3 Adsorption characteristics of FM-BC on Pb(II) in water

3.3.1 Adsorption kinetics

3.3.2 Adsorption isotherm

3.3.3 Adsorption thermodynamics

3.4 Adsorption mechanism of Pb(II) by FM-BC

Comment 7: Section 3.3.1: An increase in the specific surface area value after adsorption, as well as a decrease in potassium ions in the EDX analysis may indicate that potassium salts have washed out of the surface or pores and depending on the anions, Pb(II) could form insoluble salts. For example, PbCO3.

Response:

Thank you for your helpful comments. We have added this part in section 3.4. Please see our revised manuscript.

Section 3.4: Moreover, a decrease in potassium ions in the EDX analysis may indicate that potassium salts have washed out of the surface or pores and depending on the anions, Pb(II) probably formed more insoluble salt PbCO3.

Comment 8: And so there is still the question of the mechanism of adsorption, which may involve electrostatic attraction, hydrogen bonding, physical adsorption, ion exchange, and chemical precipitation (https://doi.org/10.1039/D0RA08055A). This discussion needs to be expanded.

Response:

Thank you for your helpful suggestions. we added relevant content in the section. Please see our revised manuscript.

Section 3.2.1: When the pH of aqueous solution was changed from 5.0 to 6.0, the contents of OH- and CO32- in the solution increased and OH- and Pb(II) formed Pb(OH)2 and Pb(OH)+, which reduced the positive charge of Pb and the content of free moving Pb(II), weakened the electrostatic attraction between FM-BC and Pb(II), and caused a minor drop in the adsorption capacity of Pb(II) by FM-BC (2.77 mg/g). It further confirmed that the Pb(II) adsorption could be related to hydrogen bonding and ion exchange.

Section 3.4: According to the XPS analysis, PbCO3 accounted for the main proportion, which indicated that chemical precipitation and complexation mainly occurred during the Pb(II) adsorption by FM-BC. The high content of Pb(NO3)2 indicated that the physical adsorption was included in the sorption process.

According to the FTIR analysis, once the FM-BC adsorbed Pb(II), the intensities of the -OH, O=C-O, and C=C peaks of alcohols or phenols decreased, indicating that O=C-O, -OH, and C=C bonds might be involved in the adsorption process. The complexation of Pb(II) with ionized oxygen functional groups (-OH or -COO) or C=C (π-electron) bonds form a complex of Pb-O and COO-Pb. In addition to that, Pb(II) can be adsorbed through an ion-exchange mechanism. as shown in the following equations:

2(R-COH) + Pb2+ →(R-CO)2Pb + 2H+

2R-COOH + Pb2+ → (R-COO)2Pb + 2H+.

According to the findings of adsorption kinetics and adsorption thermodynamics, the pseudo-second-order kinetic model displayed a more accurate description of the adsorption characteristics for Pb(II) on FM-BC, which indicated that the adsorption of Pb(II) on FM-BC was mainly governed by chemical adsorption. In general, chemisorption was dominant when △Hθ was between 20–200 kJ/mol. △Hθ of FM-BC was 32.15 kJ/mol, which indicated that chemical adsorption was the main adsorption process, agreeing with the pseudo-second-order kinetic fitting results.

Reference:

Chen, C. G.; Qiu, M. Q. High efficiency removal of Pb(II) in aqueous solution by a biochar-supported nanoscale ferrous sulfide composite. RSC Adv. 2021, 112, 953-959. https://doi.org/10.1039/d0ra08055a.

Reviewer 2 Report

The manuscript presented for review ,, Adsorption characteristics and mechanisms of Fe-Mn oxide modified biochar for Pb(II) in wastewater´´ shows interesting solution of conversion of biomass to activated carbon and then dopped with metals for adsorption application, however there are some comments needed to be properly addressed before it can be further considered

1.      The introduction should be more elaborate and include information about activated carbon, methods of obtaining and currently used biomass raw materials used for their production. The following literature may be helpful:  DOI: 10.1016 / j.jcou.2022.102071;    DOI:10.1080/03067319.2020.1717477

2.      Please emphasize the novelty of this work

3.      Page 3 point 2.1. Please add where the raw material comes from (country of origin). It is also better to replace "chille" for "cooled down"

4.      Have the authors investigated other Fe-Mn concentrations? Why was this molar ratio chosen?

5.      Figure 5 please change the images because they are of low quality and do not contain scale, especially figures 5a and 5c. In addition, in the drawings, you can probably see nanoparticles of metal, it should be more examined, it is suggested that a deeper SEM analysis be used or the TEM technique should be used,

6.      Figure 6 captions under the axes are of different sizes, please harmonize

7.      More literature comparisons are suggested

8.      improve the grammar of English

Author Response

Comment 1: The text contains typos or is poorly transformed into PDF, it should be checked.

Response:

Thank you for your helpful suggestions. We have checked the text for typos and format problems. Please see the revised manuscript.

Comment 2: Section 2.3 – “a Brunauer–Emmett– Teller analysis” – what does it mean? there is a misuse of terminology.

Response:

Thank you for your helpful suggestions. “a Brunauer–Emmett– Teller analysis” is the specific surface area analysis method, and we have rewritten this sentence. Please see our revised manuscript.

Section 2.3: The microstructure and surface properties of the samples were analyzed by scanning electron microscopy (SEM, SIGMA 300, GER) and the specific surface area of the samples were analyzed by Brunauer–Emmett–Teller method (BET) with automatic specific surface area analyzer (ASAP 2020 HD88, AUTOSORB IQ, USA).

Comment 3: Section 2.3 – “oxidation state of the samples” - there is a misuse of terminology.

Response:

Thank you for your helpful suggestions. We have rewritten the sentence. Please see our revised manuscript.

Section 2.3: The chemical composition and valence state of elements were examined using X-ray photoelectron spectroscopy (XPS, Thermo Scientific K-Alpha, USA).

Comment 4: Authors often use the phrase "reduction of Pb(II)", but this reduction has not been proven. Therefore it is better to write adsorption, uptake, or removal.

Response:

Thank you for your helpful suggestions. We have corrected the phrase "reduction of Pb(II)" in the text. Please see our revised manuscript.

Comment 5: The authors compare the surface properties of the synthesized composite with other biochars obtained from different raw materials in [30] and [32]. This is not correct at all, but in particular, because you have modified your biochar.

Response:

Thank you for the helpful comment. We have rewritten this section. Please see our revised manuscript.

Section 3.2.1: The main reason was that H+ in the aqueous solution decreased significantly with increasing pH; moreover, FM-BC had a large number of negative charges on the surface, which was beneficial for the adsorption of Pb(II) cations.

It further confirmed that the Pb(II) adsorption could be related to hydrogen bonding and ion exchange.

Comment 6: The presentation of the material is non-standard; it may be at the discretion of the authors. But the sections 3.1 and 3.2 are devoted to adsorption which is difficult to explain because the clear structure of the material was not studied, and not understood by readers yet. As a rule, the articles demonstrate the peculiarities of the adsorbent structure firstly, then the adsorption is studied and the mechanism is considered.

Response:

Thank you for the helpful comment. We changed the structure of the manuscript, first showed the peculiarities of the adsorbent structure, then studied the adsorption and considered the adsorption mechanism. Please see our revised manuscript.

3.1 The peculiarities of the adsorbent structure

3.1.1 SEM and BET analysis

3.1.2 XPS analysis

3.1.3 FTIR analysis

3.2 Impacts of pH, addition dosage, and coexisting ions on Pb(II) adsorption by FM-BC

3.2.1 The effect of pH on the adsorption of Pb(II) by FM-BC

3.2.2 The effect of FM-BC addition dosage on Pb(II) adsorption by FM-BC

3.2.3 The effect of coexisting ions on Pb(II) adsorption by FM-BC

3.3 Adsorption characteristics of FM-BC on Pb(II) in water

3.3.1 Adsorption kinetics

3.3.2 Adsorption isotherm

3.3.3 Adsorption thermodynamics

3.4 Adsorption mechanism of Pb(II) by FM-BC

Comment 7: Section 3.3.1: An increase in the specific surface area value after adsorption, as well as a decrease in potassium ions in the EDX analysis may indicate that potassium salts have washed out of the surface or pores and depending on the anions, Pb(II) could form insoluble salts. For example, PbCO3.

Response:

Thank you for your helpful comments. We have added this part in section 3.4. Please see our revised manuscript.

Section 3.4: Moreover, a decrease in potassium ions in the EDX analysis may indicate that potassium salts have washed out of the surface or pores and depending on the anions, Pb(II) probably formed more insoluble salt PbCO3.

Comment 8: And so there is still the question of the mechanism of adsorption, which may involve electrostatic attraction, hydrogen bonding, physical adsorption, ion exchange, and chemical precipitation (https://doi.org/10.1039/D0RA08055A). This discussion needs to be expanded.

Response:

Thank you for your helpful suggestions. we added relevant content in the section. Please see our revised manuscript.

Section 3.2.1: When the pH of aqueous solution was changed from 5.0 to 6.0, the contents of OH- and CO32- in the solution increased and OH- and Pb(II) formed Pb(OH)2 and Pb(OH)+, which reduced the positive charge of Pb and the content of free moving Pb(II), weakened the electrostatic attraction between FM-BC and Pb(II), and caused a minor drop in the adsorption capacity of Pb(II) by FM-BC (2.77 mg/g). It further confirmed that the Pb(II) adsorption could be related to hydrogen bonding and ion exchange.

Section 3.4: According to the XPS analysis, PbCO3 accounted for the main proportion, which indicated that chemical precipitation and complexation mainly occurred during the Pb(II) adsorption by FM-BC. The high content of Pb(NO3)2 indicated that the physical adsorption was included in the sorption process.

According to the FTIR analysis, once the FM-BC adsorbed Pb(II), the intensities of the -OH, O=C-O, and C=C peaks of alcohols or phenols decreased, indicating that O=C-O, -OH, and C=C bonds might be involved in the adsorption process. The complexation of Pb(II) with ionized oxygen functional groups (-OH or -COO) or C=C (π-electron) bonds form a complex of Pb-O and COO-Pb. In addition to that, Pb(II) can be adsorbed through an ion-exchange mechanism. as shown in the following equations:

2(R-COH) + Pb2+ →(R-CO)2Pb + 2H+

2R-COOH + Pb2+ → (R-COO)2Pb + 2H+.

According to the findings of adsorption kinetics and adsorption thermodynamics, the pseudo-second-order kinetic model displayed a more accurate description of the adsorption characteristics for Pb(II) on FM-BC, which indicated that the adsorption of Pb(II) on FM-BC was mainly governed by chemical adsorption. In general, chemisorption was dominant when △Hθ was between 20–200 kJ/mol. △Hθ of FM-BC was 32.15 kJ/mol, which indicated that chemical adsorption was the main adsorption process, agreeing with the pseudo-second-order kinetic fitting results.

Reference:

Chen, C. G.; Qiu, M. Q. High efficiency removal of Pb(II) in aqueous solution by a biochar-supported nanoscale ferrous sulfide composite. RSC Adv. 2021, 112, 953-959. https://doi.org/10.1039/d0ra08055a.

Responses to Reviewer #2

Comment 1: The introduction should be more elaborate and include information about activated carbon, methods of obtaining and currently used biomass raw materials used for their production. The following literature may be helpful:  DOI: 10.1016 / j.jcou.2022.102071;    DOI:10.1080/03067319.2020.1717477

Response: Thank your careful review. Based on this, we added relevant content in the Introduction section.

Section 1: Biochar raw materials, such as rice straw, rape straw and corn straw, which are agricultural by-products, can be pyrolyzed under anoxic and high temperature conditions to obtain raw biochar.

Reference:

Tan, G. Q.; Wu, Y.; Liu, Y.; Xiao, D. Removal of Pb(II) ions from aqueous solution by manganese oxide coated rice straw biochar - A low-cost and highly effective sorbent. J. Taiwan Inst. Chem. Eng. 2018, 84, 85-92. https://doi.org/10.1016/j.jtice.2017.12.031.

Zhao, B.; O'Connor, D.; Zhang, J. L.; Peng, T. Y.; Shen, Z. T.; Tsang, D. C. W.; Hou, D. Y. Effect of pyrolysis temperature, heating rate, and residence time on rapeseed stem derived biochar. J. Cleaner Prod. 2018, 174, 977-987. https://doi.org/10.1016/j.jclepro.2017.11.013.

Tan, G. C.; Sun, W. L.; Xu, Y. R.; Wang, H. Y.; Xu, N. Sorption of mercury (II) and atrazine by biochar, modified biochars and biochar based activated carbon in aqueous solution. Bioresour. Technol. 2016, 211, 727-735. https://doi.org/10.1016/j.biortech.2016.03.147.

Qiu, Y.; Zhang, Q.; Wang, Z. H.; Gao, B.; Fan, Z. X.; Li, M.; Hao, H. R.; Wei, X. N.; Zhong, M. Degradation of anthraquinone dye reactive blue 19 using persulfate activated with Fe/Mn modified biochar: Radical/non-radical mechanisms and fixed-bed reactor study. Sci. Total Environ. 2021, 758, 143584. https://doi.org/10.1016/j.scitotenv.2020.143584.

Yunus, Z. M.; Yashni, G.; Al-Gheethi, A.; Othman, N.; Hamdan, R.; Ruslan, N. N. Advanced methods for activated carbon from agriculture wastes; a comprehensive review. Int. J. Environ. Anal. Chem. 2022, 1021, 134-158. https://doi.org/10.1080/03067319.2020.1717477.

Comment 2: Please emphasize the novelty of this work

Response:

Thank you for your helpful suggestions. We modified the manuscript content as the follow.

Section 1: However, little research has been conducted on the adsorption features and mechanism of Pb(II) in water by the combined modification of biochar with Fe and Mn oxides. The potential role of Fe-Mn oxide biochar in removing Pb (II) from wastewater has not been developed yet.

Comment 3: Page 3 point 2.1. Please add where the raw material comes from (country of origin). It is also better to replace "chille" for "cooled down"

Response:

Thank your careful review. The country of origin of the material has been added and the "chille" has been completely replaced by "cooled down". We modified the manuscript content as the follow.

Page 3 point 2.1: Rice straw was obtained from a paddy field in Liuyang City, Hunan Province (China), and was washed and dried in an electrothermal blowing drying oven (101-1AB, Tianjin Taiste, CHN) to a constant weight before being crushed by a 100-mesh sieve to make straw powder.

Then, the biochar was cooled down to room temperature, washed with deionized water, and desiccated at 70 °C to obtain the original biochar of straw (BC).

Comment 4: Have the authors investigated other Fe-Mn concentrations? Why was this molar ratio chosen?

Response:

Thank you for the helpful comment. Through a pre-test, selected the Fe-Mn oxide biochar with the strongest adsorption capacity for Pb(II), through a pre-test, selected the Fe-Mn oxide biochar with the strongest adsorption capacity for Pb(II), Fe:Mn = 2:5. Therefore, we chose 2/5 as the experimental design. The pre-test scheme was added in the Supplementary material.

Screening of Fe/Mn molar ratio

Accurately weighed 5.0-g biochar (300°C) was infiltrated in nine kinds of Fe(NO3)3 and KMnO4 mix solutions (40 mL; Fe/Mn molar ratio =1/5, 2/5, 3/5 4/5 5/5, 5/4, 5/3, 5/2, and 5/1), respectively. After stirring for 2 h, the mixture was transferred to a 95°C water bath for 22 h, and then was heated at 300°C for 0.5 h. After cooling, the mixture was washed with deionized water several times and dried at 70°C until constant weight to obtain nine kinds of modified-biochar with different Fe/Mn molar ratio. Nine kinds of modified biochar were used for adsorption experiment, screening out the best Fe/Mn modification ratio. The best Fe/Mn molar ratio was determined based on the adsorption capacities of different biochar.

Comment 5: Figure 5 please change the images because they are of low quality and do not contain scale, especially figures 5a and 5c. In addition, in the drawings, you can probably see nanoparticles of metal, it should be more examined, it is suggested that a deeper SEM analysis be used or the TEM technique should be used.

Response:

Thank you for your helpful suggestions. We rearrangedthe images of Figure 5 to avoid occlusion; and we .Please see our revised manuscript.

Section 3.1.1: The nanoparticles of metal (Ca(II), K(I)) produced during pyrolysis were also visible from the FM-BC surface, which was more favorable for ion exchange between FM-BC and Pb(II).

Reference:

Yin, G. C.; Song, X. W.; Tao, L.; Sarkar, B.; Sarmah, A. K.; Zhang, W. X.; Lin, Q. T.; Xiao, R. B.; Liu, Q. J.; Wang, H. L. Novel Fe-Mn binary oxide-biochar as an adsorbent for removing Cd(II) from aqueous solutions. Chem. Eng. J. 2020, 389, https://doi.org/10.1016/j.cej.2020.124465.

Comment 6: Figure 6 captions under the axes are of different sizes, please harmonize.

Response:

Thank you for your helpful suggestions. We changed the structure of the manuscript and have harmonized the captions under the axes are of different sizes of Figure 2 in revised manuscript.

Comment 7: More literature comparisons are suggested.

Response:

Thank your careful review, we added more literature comparisons as follow.

Reference:

Tan, G. Q.; Wu, Y.; Liu, Y.; Xiao, D. Removal of Pb(II) ions from aqueous solution by manganese oxide coated rice straw biochar - A low-cost and highly effective sorbent. J. Taiwan Inst. Chem. Eng. 2018, 84, 85-92. https://doi.org/10.1016/j.jtice.2017.12.031.

Zhao, B.; O'Connor, D.; Zhang, J. L.; Peng, T. Y.; Shen, Z. T.; Tsang, D. C. W.; Hou, D. Y. Effect of pyrolysis temperature, heating rate, and residence time on rapeseed stem derived biochar. J. Cleaner Prod. 2018, 174, 977-987. https://doi.org/10.1016/j.jclepro.2017.11.013.

Tan, G. C.; Sun, W. L.; Xu, Y. R.; Wang, H. Y.; Xu, N. Sorption of mercury(II) and atrazine by biochar, modified biochars and biochar based activated carbon in aqueous solution. Bioresour. Technol. 2016, 211, 727-735. https://doi.org/10.1016/j.biortech.2016.03.147.

 Qiu, Y.; Zhang, Q.; Wang, Z. H.; Gao, B.; Fan, Z. X.; Li, M.; Hao, H. R.; Wei, X. N.; Zhong, M. Degradation of anthraquinone dye reactive blue 19 using persulfate activated with Fe/Mn modified biochar: Radical/non-radical mechanisms and fixed-bed reactor study. Sci. Total Environ. 2021, 758, 143584. https://doi.org/10.1016/j.scitotenv.2020.143584.

Yunus, Z. M.; Yashni, G.; Al-Gheethi, A.; Othman, N.; Hamdan, R.; Ruslan, N. N. Advanced methods for activated carbon from agriculture wastes; a comprehensive review. Int. J. Environ. Anal. Chem. 2022, 1021, 134-158. https://doi.org/10.1080/03067319.2020.1717477.

Chen, C. G.; Qiu, M. Q. High efficiency removal of Pb(II) in aqueous solution by a biochar-supported nanoscale ferrous sulfide composite. RSC Adv. 2021, 112, 953-959. https://doi.org/10.1039/d0ra08055a.

Comment 8: improve the grammar of English

Response:

Thank you for your helpful suggestions. We have already improved the grammar of English of the paper by Elsevier Language Editing Services. The certificate of Elsevier Language Editing Services is as follows:

Reviewer 3 Report

Thank you for your article that describes the preparation of iron-manganese oxide-modified biochar (FM-BC) by im-pregnating rice straw biochar with a mixture of ferric nitrate and potassium permanganate. 

The manuscript is clear, relevant for the field, and presented in a well-structured manner.  The research is scientifically sound and the experimental design is appropriate to test the hypothesis. The results are based on the details given in the Section on Proper Methods.  The conclusions are consistent with the evidence and arguments. 

Just a negligible note--better to clarify why you only looked at the effect of pH up to 6.0.

Overall impression - The article presents well-presented results of carefully chosen methods and a high-quality procedure.

Author Response

Comment: Just a negligible note--better to clarify why you only looked at the effect of pH up to 6.0?

Response:

Thank you for your helpful suggestions. We found that Pb in water began to produce obvious precipitation when pH was greater than 6.5 through our preliminary experiments. The research showed that these precipitates were Pb(OH)2. The generated precipitation seriously affected the adsorption effect of Fe-Mn modified biochar on Pb in water. Therefore, we only studied the conditions below pH 6.0 in our experiment.

Round 2

Reviewer 2 Report

- The authors write about the news of their research: ,, The potential role of Fe-Mn oxide biochar in removing Pb (II) from wastewater has not been developed yet.´´ Which is not true because earlier studies of such materials have already been published. Please check: DOI: 10.1080 / 09593330.2018.1432693

- introduction is practically untouched as suggested earlier,

- The quality of SEM images is still low, the scale is hardly visible

- the authors compare the obtained results very superficially with other studies, it is suggested to prepare a comparative table

Author Response

Comment 1: The authors write about the news of their research: The potential role of Fe-Mn oxide biochar in removing Pb (II) from wastewater has not been developed yet. Which is not true because earlier studies of such materials have already been published. Please check: DOI: 10.1080 / 09593330.2018.1432693.

Response: Thank your careful review. We deleted inaccurate sentence from the revised manuscript. Please see section 1 in our revised manuscript.

Comment 2: Introduction is practically untouched as suggested earlier,

Response:

Thank you for your helpful suggestions. We have rewritten this section, and added the source of biochar raw material, biochar preparation method and pyrolysis temperature. Please see our revised manuscript.

Section 1: Biochar has a wide range of raw materials, such as rice straw, rapeseed stem and corn straw, which are agricultural by-products. Rapid pyrolysis technology is a common method of preparing biochar from agricultural waste. Raw materials can be pyrolyzed under anoxic and at temperature range of 300–1000°C conditions to obtain raw biochar.

Comment 3: The quality of SEM images is still low, the scale is hardly visible

Response:

Thank your careful review. We replaced the low-quality graph with the high-quality one. Please see our revised manuscript.

Section 3.1.1:

(a)   BC

(b) FM-BC

(c) FM-BC-Pb

Comment 4: The authors compare the obtained results very superficially with other studies, it is suggested to prepare a comparative table.

Response:

Thank you for the helpful comment. We added a comparison table to the paper, please see our revised manuscript.

Section 3.3.1: Furthermore, the maximum adsorption size of FM-BC for Pb(II) was significantly higher than other adsorbents stated in the recent literature (Table 2).

Table 2. Comparison of sorption capacity of adsorbents for Pb(II).

Adsorbent

Max. Pb(II) capacity (mg/g)

References

Corn straw biochar

81.63

[43]

Shrimp shell biochar

94.16

[44]

Calcined mussel shell powder

102.04

[45]

CuFe2O4-loaded corncob biochar

132.10

[46]

BC

70.33

This study

FM-BC

165.88

Wang, L. C.; Zhao, H. H.; Song, X. L.; Li, Y. K.; Li, D. The adsorption characteristics and mechanism of Pb(II) onto corn straw biochar. J. Biobased Mater. Bioenergy. 2021, 153, 287-295. https://doi.org/10.1166/jbmb.2021.2056.

Feng, T.; Yi, T.; Wang, Q. B.; Li, P. W. Shrimp shells-derived biochar for efficient adsorption of Pb2+ in aqueous solutions. Desalin. Water Treat. 2021, 233, 106-117. https://doi.org/10.5004/dwt.2021.27558.

Wang, Q.; Jiang, F. Y.; Ouyang, X. K.; Yang, L. Y.; Wang, Y. G. Adsorption of Pb(II) from Aqueous Solution by Mussel Shell-Based Adsorbent: Preparation, Characterization, and Adsorption Performance. Mater. 2021, 144, https://doi.org/10.3390/ma14040741.

Zhao, T. C.; Ma, X. L.; Cai, H.; Ma, Z. C.; Liang, H. F. Study on the Adsorption of CuFe2O4-Loaded Corncob Biochar for Pb(II). Molecules. 2020, 2515, https://doi.org/10.3390/molecules25153456
